# Acoustic Lung Imaging Utilized in Continual Assessment of Patients with Obstructed Airway: A Systematic Review

**DOI:** 10.3390/s23136222

**Published:** 2023-07-07

**Authors:** Chang-Sheng Lee, Minghui Li, Yaolong Lou, Qammer H. Abbasi, Muhammad Ali Imran

**Affiliations:** 1James Watt School of Engineering, University of Glasgow, Glasgow G12 8QQ, UK; leechangsheng@outlook.com (C.-S.L.); qammer.abbasi@glasgow.ac.uk (Q.H.A.); muhammad.imran@glasgow.ac.uk (M.A.I.); 2Global Technology and Innovation Department, Hill-Rom Services Pte Ltd., Singapore 768923, Singapore; lou.yaolong@gmail.com

**Keywords:** acoustic lung imaging, frequent lung assessment, integrated devices, lung function application, obstructed airway identification, sensing and imaging

## Abstract

Smart respiratory therapy is enabled by continual assessment of lung functions. This systematic review provides an overview of the suitability of equipment-to-patient acoustic imaging in continual assessment of lung conditions. The literature search was conducted using Scopus, PubMed, ScienceDirect, Web of Science, SciELO Preprints, and Google Scholar. Fifteen studies remained for additional examination after the screening process. Two imaging modalities, lung ultrasound (LUS) and vibration imaging response (VRI), were identified. The most common outcome obtained from eleven studies was positive observations of changes to the geographical lung area, sound energy, or both, while positive observation of lung consolidation was reported in the remaining four studies. Two different modalities of lung assessment were used in eight studies, with one study comparing VRI against chest X-ray, one study comparing VRI with LUS, two studies comparing LUS to chest X-ray, and four studies comparing LUS in contrast to computed tomography. Our findings indicate that the acoustic imaging approach could assess and provide regional information on lung function. No technology has been shown to be better than another for measuring obstructed airways; hence, more research is required on acoustic imaging in detecting obstructed airways regionally in the application of enabling smart therapy.

## 1. Introduction

Respiratory diseases such as asthma, bronchitis, chronic obstructive pulmonary disease (COPD), coronavirus-2 disease 2019 (COVID-19), and pneumonia are the most common cause of obstruction in airways that affect the lung, leading to chest pain, shortness of breath, coughing, and mucus production [1,2,3,4].

Respiratory therapy enhances the mobilization of mucus in the lung to reduce airway resistance and improve breathing [5]. For example, a high-frequency chest wall oscillation device is used to lower the mucus viscosity through percussion and vibration on the chest and creates the coughing action, which helps to eject the mucus from the airway [5,6,7]. High-frequency chest wall oscillation devices have been enhanced over the years, such as integrating electronic control for specified pressure oscillating discs to deliver palpitation directly to targeted chest areas [6,7]. The patient’s current level of lung function greatly influences the oscillating disc’s intensity and length of therapy. Before making a change to the respiratory therapy parameter, patients must report quarterly to the hospital for evaluation of their lung function. Thus, frequent on-demand regional assessment of lung function is imperative to enable smart respiratory therapy and delivery of efficient treatment, such as only targeting identified affected airways and adjusting the therapy parameters promptly to optimize and reduce the duration for patients with respiratory diseases.

Traditional lung function assessment, such as chest X-rays, computed tomography (CT), and magnetic resonance imaging (MRI), has the advantages of high-resolution imaging, but includes patient-to-equipment approaches, unsuitable for frequent assessment due to the ionizing radiation effect on the patient’s health, and poses the risk of transporting patients to the equipment in the clinical setting [8,9]. There is a lack of equipment accessibility, especially in small communities during the recent COVID-19 outbreak, where movement restrictions added to the disadvantage of the patient-to-equipment approach [10]. The advances in the nonionizing acoustic approach to the lung function assessment have enabled the equipment-to-patient (bedside/portable) approach and frequent lung function assessment [11,12,13,14], where obstructed airways affect sound transmission (acoustic signals) routes and have spectral and regional impacts that can benefit from several measurements over the chest area [13,14,15]. A brief qualitative comparison between traditional lung function assessment and acoustic imaging is presented in Table 1.

To date, reviews on lung function assessment have a broad focus [11,13,16,17,18,19,20,21]. For example, Cammarota et al. [11] reviewed various advanced equipment-to-patient bedside monitoring techniques through esophageal pressure, diaphragm’s electrical activity, and monitoring tools such as electrical impedance tomography and ultrasound on patients with acute respiratory failure. Rao et al. [13] reviewed different types and approaches of acoustic outcome measures on lung functions. Kolodziej et al. [16], Ramsey et al. [17], and Dubsky et al. [18] reviewed patient-to-equipment approaches that require patient preparation and mainly non-acoustic approaches. Lauwers et al. [19] reviewed multidisciplinary outcome measures that were utilized to evaluate the respiratory therapy’s effectiveness in participants below the age of eighteen with COPD. Augustin et al. [20] and Oliveira et al. [21] concentrated on the patient-based reported outcome, such as the patient’s quality of life. To the best of our knowledge, previous studies on the potential of bedside/portable acoustic imaging to achieve similar outcome measures as those patient-to-equipment approaches have not been systematically reviewed. Hence, this systematic review aimed to answer the following research question: can equipment-to-patient acoustic imaging be used as a continuing lung function assessment tool, enabling smart therapy for patients with respiratory diseases?

## 2. Methods

This systematic review was reported according to the updated Preferred Reporting Items for Systematic Reviews and Meta-Analyses 2020 (PRISMA) guidelines [22] (Table A1, Appendix A). The systematic review was conducted as per the registered PROSPERO protocol record (CRD42023417131), https://www.crd.york.ac.uk/PROSPERO/display_record.php?RecordID=417131, accessed on 18 May 2023.

### 2.1. Search Strategy and Study Selection

The search strategies were constructed a priori using different terms relating to continuing beside/portable acoustic imaging on regional lung health/function. A thorough description of the search strategy and terms are shown in Table A2, Appendix B. The literature search was performed between 31 March 2023 and 14 April 2023. The terms used in the search were defined based on the critical elements from the SPIDER (Sample, Phenomenon of Interest, Design, Evaluation, Research Type) model, as the SPIDER model is suitable for a qualitative evidence-based systematic review [23,24]. Suitable keywords were selected, e.g., patients with obstructed airway or chronic respiratory diseases (CRD) refer to S (sample), PI (the phenomenon of interest) relates to the bedside/portable acoustic images, D (design) is the published literature of any research design, E (evaluation) is referenced to the assessment tool characteristics, and R (research type) connects to qualitative, or quantitative, or both research study types. To be as inclusive as possible and in addition, our review questions did not have a specific study methodology; hence, D (design) and R (research type) elements of SPIDER were excluded from keyword selection. The search was conducted by one reviewer (C.-S.L.) from the following reference databases published in English: Web of Science, PubMed, ScienceDirect, Scopus electronic database, Google Scholar, and SciELO Preprints. There were no restrictions on publication date or participants’ age, while conference proceedings and studies on animals were excluded. Thirty percent of the identified records from the database, randomly selected, were evaluated by a second reviewer (Y.L.). Four disagreements were discussed and resolved with consensus between the two reviewers.

### 2.2. Data Collection and Synthesis

Titles and abstracts were screened at the first stage. In the second stage, the introduction was reviewed to ensure that the selected studies’ objectives fit the research question. A full-text review of possible potential papers from the shortlisted studies’ reference list was performed in the third stage. The corresponding and first author of the shortlisted papers were used to avoid introducing bias, double counting, and possible duplicate publications from the same group. Two reviewers (C.-S.L. and Y.L.) extracted relevant data from the studies included in the qualitative synthesis and review using a customized spreadsheet containing study variables: author and the year of publication, study design, study population, technique, measured respiratory disease, recording venue, and significant outcome. There was complete agreement between the two reviewers in terms of data extraction. Key outcomes in this systematic review refer to the individual-identified studies reporting statistically significant ability to perform home-based or bedside assessment of lung function, regardless of the statistical analysis used.

Meta-analyses were not performed as the studies were conducted in various populations and used varied definitions and statistical analyses on the measure of lung health/function outcomes; therefore, appraisals and findings of each study were given independently.

### 2.3. Risk of Bias

Different risk-of-bias tools exist for different study types, such as Cochrane risk of bias (RoB) for randomized trials and National Institutes of Health quality assessment tools for controlled intervention studies [25]. An adapted form of the Newcastle–Ottawa Scale (NOS) [25,26,27] for cross-sectional studies was utilized in this review as the studies selected for quality review and synthesis were purely cross-sectional studies. Each reviewer graded each item based on the information provided in the articles. Item 1 has a maximum of five stars; it was graded with stars if the sample size truly represented the average target population or was somewhat representative of the average in the target group, sample size was justified, the response rate was satisfactory, and it was measured with a validated measurement tool. Item 2 has a maximum of two stars and was graded with stars if the confounding factors were controlled and the study controlled for any additional factor. Item 3 has a maximum of three stars; two stars were graded if the outcome was assessed by independent blind assessment or record linkage, and one star for statistical analysis used, including indication of confidence intervals. Each study’s quality score was determined as the sum of all scores, ranging from 0 to 10 points, with higher scores indicating higher quality. In addition, for a fair review, no weighting was applied as any possible area for bias to be more crucial than another was considered. Two reviewers (C.-S.L. and Y.L.) independently assessed the risk of bias with the NOS tool for each included study. There was no disagreement between the two reviewers with regard to the quality assessment of the included studies. The details of the quality assessment are presented in the Table A3, Appendix C.

## 3. Results

### 3.1. Study Selection

A total of five hundred and ten papers were identified from databases and registers, where the electronic database search yielded four hundred and ninety-six papers and fourteen papers were identified from cross-reference and citation. One hundred and fifty-three records were screened after three hundred and fifty-seven duplicates were eliminated, of which seventeen were assessed in full text. After the review process presented in Figure 1, two papers were excluded due to the unavailability of the full text. The study selection process led to fifteen studies, which were included in the review for quality assessment and synthesis of results.

### 3.2. Study Characteristics

Table 2 narrates each of the selected studies in terms of study design, study population, approach, diseases, venue of the assessment, and the primary outcome of the techniques in terms of lung health. These fifteen studies from Table 2 were conducted on patients with obstructed airways relating to respiratory diseases and were cross-sectional studies. Five studies [28,29,30,31,32] focused on patients with CRD, while six studies [33,34,35,36,37,38] reported on patients with lung consolidation, and the remaining four studies [39,40,41,42] investigated on COVID-19 patients. Eight studies [28,32,33,35,39,40,41,42] utilized lung ultrasound (LUS), and the remaining seven studies [29,30,31,34,36,37,38] utilized vibration response imaging (VRI) technology as observed in Table 2. The total number of participants per study ranged between ten and two hundred and nineteen. The study population inclines slightly towards males (684/1190) at about 57%. Three studies [33,35,41] experimented on children below the median age of thirteen, four studies [28,29,36,39] investigated on elderly above the median age of sixty, and the remaining eight studies [30,31,32,34,37,38,40,42] tested on adults between the median age of thirteen and sixty.

Positive observation of either change in geographical lung area or sound energy, or both, was the most common outcome obtained by eleven studies [29,30,31,32,34,37,38,39,40,41,42]. The remaining four studies [28,33,35,36] reported positive observations of lung consolidations. Twelve studies [28,29,30,31,32,33,35,37,39,40,41,42] performed the assessment in an uncontrolled environment such as hospitals, ICUs, or clinics, while, the remaining three studies [34,36,38] performed the assessment in a controlled setting. Eight studies [28,32,35,36,37,39,40,42] compared lung assessment from two different techniques. One study [37] compared VRI with chest X-ray, one study [36] compared VRI with LUS, two studies [28], [35] compared LUS with chest X-ray, and the remaining four studies [32,39,40,42] compared LUS against CT.

### 3.3. Quality Scores in Individual Studies

The quality assessment of individual studies is summarized in Table 3. The fifteen selected studies scored in the range of five to nine using an adapted form of NOS for cross-sectional studies, where studies were classified as unsatisfactory studies (0–4 points), satisfactory studies (5–6 points), good studies (7–8 points), and very good studies (9–10 points) [27,43]. From Table 3, two studies [28,42] were identified as very good studies, nine studies [30,32,33,34,35,36,37,39,40] were determined as good studies, while the remaining four studies [29,31,38,41] were satisfactory. None of the selected studies was unsatisfactory. The justification, such as calculation or the derivation of the sample size, was not provided in all fifteen shortlisted studies; hence, no rating was given in the sample size column. Although statistical tests and analyses were used in all fifteen shortlisted studies, no rating was given to [29,31,38] in the statistical column as the confidence intervals were not explicitly mentioned in the studies.

### 3.4. Results of Individuals/Synthesis

From the synthesis of the shortlisted studies in Table 2, LUS [28,32,33,35,39,40,41,42] and VRI [29,30,31,34,36,37,38] are the main techniques that utilized acoustic signals and translated the signals into imaging for frequent bedside/portable assessment. The overview working principles of LUS and VRI are presented in Figure 2. The synthesis results are presented according to the type of outcome measure, with further divisions made based on the compared factors and technologies.

#### 3.4.1. Lung Ultrasound

Lung ultrasound images are based on sound propagation in matter and sound wave interaction with reflecting interfaces [44]. LUS has made significant progress in evaluating lung pathologies in the last two decades and is noninvasive, nonionizing, and safe to repeat the lung function assessment at the patient’s bedside numerous times, leading to the reduction in chest X-rays and CT examinations [14,44,45]. LUS has been proposed as an on-demand examining tool to avoid intra-hospital transport, where intra-hospital transport of patients requires accompanying costs, such as planning resources and personal organization, and to reduce the risk of patient cross-contamination and radiation exposure [14]. The impact of lung ultrasound on economics is detailed in [14]. LUS has also been considered in emergency settings, such as pulmonology and thoracic surgery ambulatory clinics [44,45].

#### 3.4.2. Lung Ultrasound against Chest X-rays

Lung ultrasound demonstrated statistical equivalence to chest X-rays in detecting respiratory diseases such as lung consolidation and pleural effusion from fifty patients’ results, in terms of sensitivity [35]. The radiologic chest X-ray score of extravascular lung consolidation had a substantial linear connection with the LUS echo comet score from one hundred and thirty-five images [28]. A significant correlation was found with regard to lung consolidation when the radiologic chest X-ray score was compared with the LUS echo comet score of a single chest intercostal space, specifically on the right side at the third intercostal space on the anterior axillary line [28]. Hence, compared to chest X-rays, LUS can demonstrate statistically equal sensitivity for respiratory findings, such as CRD, pleural effusion, and lung consolidation [28,35].

#### 3.4.3. Lung Ultrasound against Computed Tomography

The sensitivity and specificity of LUS for each patient’s distinct lung zones were evaluated using chest CT findings as a reference, as CT is the gold standard for evaluating pulmonary abnormalities [32,39,40,42]. The LUS data from two hundred and nineteen patients achieved an overall sensitivity and specificity of 75% (1348/1801) and 66% (549/827), respectively, with CT findings as a reference [42]. LUS was able to identify the differences in the airway wall thickness, statistically comparable to CT from sixty patients’ data, and provide better visualization when compared against the healthy group [32]. The LUS score and CT had a strong correlation, where thirty-seven patients (72.5%) from CT scans were suggestive of COVID-19 or had radiologic symptoms, while LUS exams suggested forty patients (78.4%) [39]. With a sensitivity of 100%, specificity of 78.6%, positive predictive value of 92.5%, and negative predictive value of 100%, LUS accurately diagnosed all thirty-seven patients with abnormal findings on CT [39]. With LUS compared to CT, there were no missed diagnoses of COVID-19 in the group [39]. Similar to [39], when compared to CT, LUS demonstrated statistical equivalence in detecting COVID-19 and lung abnormalities from twelve patients’ data [40].

#### 3.4.4. Vibration Response Imaging

Vibration response imaging has been proposed to monitor respiratory distribution within the lungs dynamically and is regarded as an electronic stethoscope alternative that records vibrations emitted from the chest using an array of microphones and converts them into grey-scale images [31,46,47]. The hypothesis is that when there are changes in airflow in the lungs, frequency, and intensity, these changes will affect the lung vibration response images [31,46,47]. The contact sensors on the posterior of the patient’s chest wall will simultaneously record 12 to 20 s sound clips. The recordings are converted into digital signals and filtered through a bandpass filter to minimize artifacts such as sounds produced by the environment and heart. The filtered output signal combined with an interpolating function is expressed as an image of breath sound intensities between measured locations and the microphone’s location on the chest wall concerning time [46]. VRI images are scored based on the image quality; intensity of the vibrational energy curve; abnormal signs in the image output: unsmooth, inspiratory steep, spike, or step dynamic image; image movement during breathing phases; and maximal energy frame shape. Thus, VRI technology was an excellent way to detect lung sound distribution during mechanical ventilation in several studies [31,47,48].

#### 3.4.5. Vibration Response Imaging against Chest X-rays

Four individuals with pneumonia but no consolidation had lower vibration intensity than thirteen patients with pneumonia plus consolidation (8 ± 14 vs. 22 ± 29 × 10^6^ AU) [37]. The consolidation identified by chest X-rays overlaps with the increased vibration intensity area, which is represented by darker colors in VRI [37]. This great intensity overshadows the appearance of the left lung due to normalization [37]. The vibration intensity difference between freely breathing and mechanically ventilated patients was significant [37].

#### 3.4.6. Vibration Response Imaging against Lung Ultrasound

In the per-patient study (forty-five cases), VRI can accurately (45/56, 80%) identify the proper diagnosis (right, left, or bilateral effusion) [36]. In the per-hemithorax study, the agreement between the VRI recording and the chest X-rays on the amount of effusion was 74% (83/112) [36].

## 4. Discussion

While exposing patients to unnecessary radiation doses and straining medical resources should be circumvented, clinicians and doctors should consider the assessment of the respiratory system by equipment-to-patient acoustic imaging. A detailed understanding is needed, i.e., a potential indication of the continual assessment of patients with obstructed airways through acoustic imaging, which can lead to optimal respiratory therapy. Therefore, this systematic review aims to address the capability of acoustic imaging as a home-based and continuous outcome assessment of lung function for patients with obstructed airways. This systematic review identified LUS (8/15 studies) and VRI (7/15 studies) as the implemented approaches for home-based/bedside imaging of patients with an obstructed airway. Compared to chest X-rays, LUS and VRI have demonstrated similar accuracy in diagnosing respiratory diseases, particularly pleural effusion and lung consolidation in critically ill patients [28,35,37,45]. Compared to CT, LUS has demonstrated similar reliability in the assessment of (n = 58) COVID-19 patients [39,40,49] and has presented the potential to detect changes in the airway thickness in (n = 60) patients with obstructed airways when compared to a healthy group [32]. Although there is no comparison between VRI and CT in our shortlisted studies, VRI can accurately (45/56, 80%) identify the correct obstructed region, e.g., right, left or bilateral effusions, when compared to LUS [36]. Table 4 offers an overview of the critical factors for the discussed outcome measures.

Since each outcome measure has advantages and unique problems, no perfect approach or one approach that is superior to another could be found. In general, this systematic review summarizes a diverse range of investigations conducted over the last two decades. The use of certain outcome measures, study designs, etc., varied widely. Hence, our goal is not to make any claims about ideal acoustic lung imaging but to examine the applicability of various acoustic lung imaging in patients with obstructed airways. Relevant studies were searched in six databases that included a wide range of research articles and a lengthy period, as no restriction on publication timeline for each database was applied to access as much pertinent literature as possible. Precise inclusion and exclusion criteria via SPIDER were utilized regarding the population, exposures, and study outcome. The overall research risk of bias quality was evaluated with an adapted form of NOS.

One potential drawback is that our search may have missed certain significant studies due to the language barrier, as only journals that published studies in English were considered. As only two main techniques, LUS and VRI, were reported in this systematic review, and there is limited published research on acoustic lung imaging for patients with obstructed airways, conclusive statements about the ideal technology for the population could not be made. Additionally, as the majority of the included research exclusively addressed continuing lung assessment with acoustic lung imaging on patients with obstructed airways, other patients with tumors, cancer, or a combination of obstructed airways and tumors may find our findings less helpful. Because convenient sampling was used to gather the data for this study, it is important to interpret the results carefully. Lastly, this study was not a controlled, randomized experiment. As a result, the reported imaging effects indicated observations and trends in the assessment of lung function.

Two interdependent main areas of interest can be pointed out for future research. The first is the evaluation of acoustic imaging for regional lung assessment patients with obstructed airways. A common trait that can be observed is that LUS and VRI compute the impedance or the resistivity in the lung or the airway as an indicator for lung function assessment and then convert the signal data to an intuitive image or medical image. With the information on the regional lung information, doctors and clinicians could enhance the ACT with timely adjustment. From the fifteen selected studies in Table 2, LUS and VRI imaging could be a sensitive measure to quantify local and regional changes in lung pathology. Lung sound and vibration energy produced from the chest wall could be transformed into information that presents local ventilation status and could increase future knowledge of airway therapy’s effectiveness. Second, acoustic lung imaging modeling and simulation have not been explored, and the understanding of the sensor’s placement, position, and effect on the outcome measure has not been investigated. These methods could reveal important details about the physiological processes that underlie targeted therapies, revealing distinctions between various therapeutic modalities. Computerized lung sound monitoring could be a sensitive approach to evaluate regional changes in the airways brought on by mucus displacement and better regional ventilation.

## 5. Conclusions

This systematic review has described the potential and limitations of bedside/portable acoustic imaging, such as LUS and VRI, in the continual and frequent assessment of lung function. LUS and VRI have shown the potential to achieve similar results as the traditional imaging modality with the small number of selected studies in this systematic review. There is no direct superiority, e.g., LUS is better than VRI or vice versa, as each acoustic imaging technique has unique advantages and limitations for measuring the obstructed airway regionally and frequently. Further acoustic imaging research, especially on converting lung sound into images for assessment in VRI, is required. VRI requires a controlled environment and is deemed not as established as LUS, whereas LUS has been tested in the hospital and ICU setting and used in the pilot/comparison study to identify obstructed airways in COVID-19. VRI has the potential for home-based usage as no medical interpretation of the results is required, unlike LUS, which requires medical interpretation of the results. In theory, acoustic imaging is valuable and sensitive for identifying obstructed airway regions instead of diagnosing respiratory disease.

## Figures and Tables

**Figure 1 sensors-23-06222-f001:**
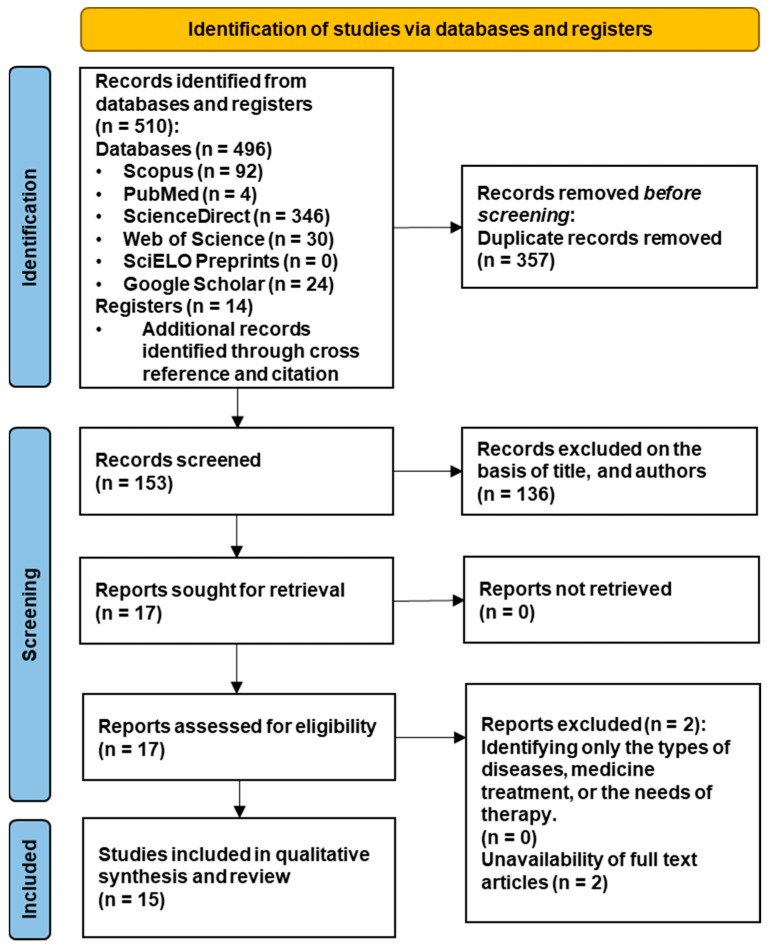
PRISMA 2020 flow of information for study selection and inclusion.

**Figure 2 sensors-23-06222-f002:**
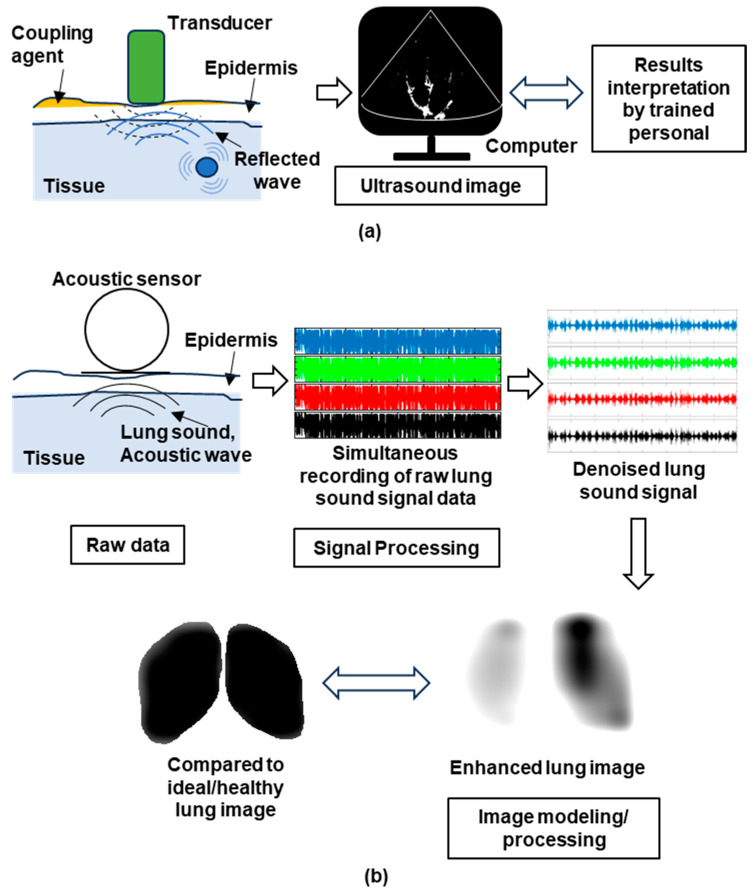
The conceptual flow of acoustic imaging system working principle. (**a**) Lung ultrasound and (**b**) vibration response imaging.

**Table 1 sensors-23-06222-t001:** Advantages and limitations between traditional- and acoustic-lung imaging in assessing lung function.

	Traditional Lung Function Assessment (Chest X-ray, CT, MRI)	Acoustic Imaging Lung Assessment
Benefits	Typically outputs high image resolutions Planar lung imaging, two- and three-dimensional image assessment Established approach for diagnostic purposes Typically results have high sensitivity and specificity	Typically portable and accessible, hospital/equipment-to-patient approach Lower operation costs, reducing time in preparing patients for assessment Planar lung imaging, fast assessment time Reduced disease cross-contamination risk of transporting patients, particularly in a hospital setting Established approach for early analysis of lung function Frequent lung function assessment due to nonionizing approach
Limitations	Moderate accessibility, patient-to-equipment/hospital approach High operating cost, requiring patient preparation and planning or assessment Radiation factor Risk of cross-contamination of diseases in a hospital setting through patient transport to the equipment	Low image resolution Typically for assessment, unsuitable for diagnostic purposes Limited surface visualization Lower sensitivity compared to traditional lung function assessment

**Table 2 sensors-23-06222-t002:** The findings of the fifteen shortlisted studies’ characteristics.

Author (Year)	Study Design	Sample Population	Technique	Venue	Diseases	Outcome
Jambrik et al. [28] (2004)	Cross-sectional	*n* = 121 Female: 43 Male: 78 Age: 67 ± 12	LUS	ICU	Chronic pulmonary disease	Pulmonary abnormalities
Dellinger et al. [29] (2007)	Cross-sectional	*n* = 38 Female: 24 Male: 14 Age: 60 ± 16	VRI	ICU	Chronic pulmonary disease	Geographical lung area and sound energy change
Anantham et al. [36] (2009)	Cross-sectional	*n* = 56 Female: 23 Male: 33 Age: 68 ± 13	VRI	Controlled environment	Pleural effusion	Bilateral effusion
Guntupalli et al. [30] (2009)	Cross-sectional	*n* = 66 Female: 32 Male: 34 Age: 56 (Median)	VRI	Hospital	Asthma, COPD	Geographical lung area and sound energy change
Lev et al. [37] (2010)	Cross-sectional	*n* = 82 Female: 57 Male: 25 Age: 59 ± 19	VRI	ICU	Consolidation, congestion, pleural effusion, atelectasis	Geographical lung area and sound energy change
Bing et al. [31] (2012)	Cross-sectional	*n* = 36 Female: 12 Male: 24 Age: 58.34 ± 14.72	VRI	Outpatient clinic and ICU	Acute exacerbation of COPD	Geographical lung area and sound energy change
Liu et al. [38] (2014)	Cross-sectional	*n* = 23 Female: 10 Male: 13 Aged: 56 ± 2	VRI	Controlled environment	Idiopathic pulmonary fibrosis	Geographical lung area change
Ambroggio et al. [35] (2016)	Cross-sectional	*n* = 132 Female: 58 Male: 74 Age: 4.4 (Median)	LUS	Hospital	Pneumonia, wheezing, bronchiolitis, pleural effusion	Lung consolidation
Gorska et al. [32] (2016)	Cross-sectional	*n* = 60 Female: 28 Male: 32 Age: 31-72	LUS	Outpatient clinic	Asthma, COPD	Geographical bronchial wall thickness change
Jiang et al. [34] (2017)	Cross-sectional	*n* = 62 Female: 28 Male: 34 Age: 43.12 ± 13.64	VRI	Controlled environment	Pneumonia	Sound energy change
Chen et al. [39] (2020)	Cross-sectional	*n* = 51 Female: 23 Male: 28 Aged: 61 (Median)	LUS	Hospital	COVID-19, other respiratory symptoms	Geographical lung intensity change
Giorno et al. [40] (2020)	Cross-sectional	*n* = 34 Female: 13 Male: 21 Age: 13 (Median)	LUS	Hospital	COVID-19	Geographical lung intensity change
Musolino et al. [41] (2020)	Cross-sectional	*n* = 10 Female: 4 Male: 6 Age: 11 (Median)	LUS	Hospital	COVID-19	Geographical lung intensity change
Ruiz et al. [33] (2020)	Cross-sectional	*n* = 200 Female: 84 Male: 116 Age: 0.4 (Median)	LUS	Hospital	Bronchiolitis	Pulmonary abnormalities
Rizzetto et al. [42] (2021)	Cross-sectional	*n* = 219 Female: 67 Male: 152 Age: 58 (Median)	LUS	Hospital	COVID-19	Geographical lung intensity change

Abbreviations: ICU, intensive care unit; LUS, lung ultrasound; *n*, total sample population.

**Table 3 sensors-23-06222-t003:** Quality assessment of the included fifteen studies using Newcastle–Ottawa scale adapted for cross-sectional studies.

	Selection (5)				Comparability (2)	Outcome (3)		
Study ID	Representativeness of the Sample	Sample Size	Non-Respondents	Ascertainment of the Exposure (Risk Factor)	Comparability of Subjects in Different Outcome Groups on the Basis of Design or Analysis	Assessment of Outcome	Statistical Test	Total (10)
Jambrik et al. [28] (2004)	★		★	★★	★★	★★	★	9
Dellinger et al. [29] (2007)	★		★	★	★	★		5
Anantham et al. [36] (2009)	★		★	★	★★	★	★	7
Guntupalli et al. [30] (2009)	★		★	★	★	★★	★	7
Lev et al. [37] (2010)	★		★	★	★★	★	★	7
Bing et al. [31] (2012)	★		★	★	★	★★		6
Liu et al. [38] (2014)	★		★	★	★	★		5
Ambroggio et al. [35] (2016)	★		★	★★	★	★★	★	8
Gorska et al. [32] (2016)	★		★	★★	★	★★	★	8
Jiang et al. [34] (2017)	★		★	★	★	★★	★	7
Chen et al. [39] (2020)	★		★	★★	★	★★	★	8
Giorno et al. [40] (2020)	★		★	★	★	★★	★	7
Musolino et al. [41] (2020)	★		★	★	★	★	★	6
Ruiz et al. [33] (2020)	★		★	★★	★	★★	★	8
Rizzetto et al. [42] (2021)	★		★	★★	★★	★★	★	9

NB, the numbers in parentheses are maximum scores to be given per category. ★ denotes the total score given for each sub-category.

**Table 4 sensors-23-06222-t004:** Key considerations for continuing assessment of lung function.

	LUS	VRI
Approach	Detects the sound wave interaction with reflecting interfaces such as the lung tissue via a specialized probe	Measures breathing sound distribution in the airway and converts it to vibration energy with an electronic stethoscope/microphone
Imaging	Maps from the sound propagation that is reflected from the lung tissue or rib cage	Maps the ventilation distribution into a grayscale figure for lung function assessment
Indications	Assesses lung health regionally and globally Flexible, bedside, and home-based monitoring are possible Frequent, semi-continuous monitoring due to a nonhazardous approach Comparable assessment outcome of lung function compared to CT and chest X-rays	Assesses lung health regionally and globally Flexible, bedside, and home-based monitoring are possible. Frequent, semi-continuous monitoring due to a nonhazardous approach Maps the vibration energy with one planar posterior measurement Good correlation of lung function assessment compared to LUS and chest X-rays
Disadvantages	Requires specialized training to operate the equipment Requires trained personnel to interpret the assessment outcome Assessment outcome may be affected by the patient’s body size [44]	Requires a controlled environment and additional equipment, such as a vacuum pump Expensive system at USD 50,000 [50] as compared to a typical LUS system at about USD 33,000 [14]

## Data Availability

No new data were created or analyzed in this study. Data sharing is not applicable to this article.

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
