# Peer review of "Acoustic Lung Imaging Utilized in Continual Assessment of Patients with Obstructed Airway: A Systematic Review"

_sensors, 2023, doi:10.3390/s23136222_

Round 1

Reviewer 1 Report

I read your review with attention. The work is interesting and sounds good. I invited the authors to discuss the economic impact and to think that not all is not gold for the future.

Ref: 

1) Cammarota G, et al. Advanced Point-of-care Bedside Monitoring for Acute Respiratory Failure. Anesthesiology. 2023 Mar 1;138(3):317-334. doi: 10.1097/ALN.0000000000004480. PMID: 36749422.

2) Cammarota G, et al. Lung ultrasound monitoring: impact on economics and outcomes. Curr Opin Anaesthesiol. 2023 Apr 1;36(2):234-239. doi: 10.1097/ACO.0000000000001231. Epub 2022 Dec 28. PMID: 36728722.

Acceptable

Reviewer 2 Report

The review provides a clear introduction to respiratory diseases and their impact on lung function. Moreover, it highlights the limitations of traditional approaches for lung function assessment and emphasizes the potential of acoustic imaging as a nonionizing and portable approach for frequent lung function assessment. This review is based on the organization of previous reviews on lung function assessment, pointing out the lack of systematic reviews on bedside/portable acoustic imaging. The review has a clear logic line and is straightforward which can help the readers have a better understanding of the topic. However, some minor revisions are still needed before it can be published. The detailed comments are as follows.

1. What are the benefits, risks, and limitations of traditional lung function assessment methods such as chest X-ray, CT, and MRI? I suggest a brief comparison (table or figures) between traditional assessment methods and acoustic imaging, including accessibility, portability, sensitivity, specificity, testing time and cost, etc. 

2. Elaborate working principles of specific outcome measures used in previous studies on lung function assessment should be added. I suggest the author(s) think about adding some schematic illustrations or flowcharts for these methods to make the review more comprehensive.

3. Can the author(s) clarifies how can bedside/portable acoustic imaging be used as a tool for smart therapy in respiratory diseases? Is it aimed at wearable continuous monitoring tools or an alternative method for medical diagnosis? Does the author(s) believe that acoustic imaging will have enough sensitivity during these scenarios?

4. What are the specific benefits of frequent regional assessment of lung function in optimizing respiratory therapy for patients with respiratory diseases? It is necessary to emphasize the frequent/long-term monitoring of the lung function? 

Minor editing of English language required.

Reviewer 3 Report

It is imperative to enable smart respiratory therapy and delivery of efficient treatment by frequent regional assessment of lung function. This systematic review described an overview of the potential and limitations of equipment-to-patient acoustic imaging, such as LUS and VRI, in the continual and frequent assessment of lung function. Both LUS and VRI have demonstrated statistically equivalent in achieving similar results as the traditional imaging modality with the limited number of selected studies.

The manuscript is well-written and well-organized. The objective is well-articulated and reached. The figures and tables are presented in a clear and appropriate manner and are consistent with the description in the text. The results and analysis presented in the manuscript are interesting for this field and Sensors is the appropriate journal to submit it. There are a few points that the authors may want to consider, as described in the following.

In line 17-18, “2 studies comparing LUS to computed tomography (CT) and 4 studies comparing LUS in contrast to CT.” According to the results shown in section 3.4, LUS was compared to Chest X-rays and CT, respectively.

In Table 2, the column of “Sample Size” is empty. And some elements in the column of statistical test are empty, too. Can the authors explain why?

Reviewer 4 Report

General comment

This systematic review provided an overview of the suitability of equipment-to-patient acoustic imaging in continual assessment of lung conditions. The literature search was conducted using Scopus, Pub-Med, ScienceDirect, Web of Science, SciELO Preprints, and Google Scholar.  Fifteen studies remained for additional examination after the screening process. Two imaging modalities, lung ultrasound  and vibration imaging response, were identified. This study findings indicated that the acoustic imaging approach could perform an assessment and provide regional information on lung function. But, no technology has been shown to be better than another for measuring obstructed airways.

Specific points

1-The authors should not use numbers at the beginning of sentences and lines. I think, the authors should indicate with writing the numbers at the beginning of the sentence (e.g.) Fifteen, One hundred fifty three, Five, Four

-15 studies remained for additional examination after the screening process.

-153 records were screened after 357 duplicates were eliminated, of which 17 were assessed in full text.

-5 studies [26], [27], [28], [29], [30] focused on patients with CRD, while 6 studies [31], [32], [33], [34], [35], [36] reported on patients with lung consolidation, and the remaining 4 studies [37], [38], [39], [40] investigated on COVID-19 patients

-4 individuals with pneumonia but no consolidation had lower vibration intensity than 13 patients with pneumonia plus consolidation (8 ± 14 vs 22 ± 29 x 10^6 AU) [35].

2-I think, the authors should indicate in writing the numbers less than ten in manuscript (e.g.) one, two, four, eight…

- The most common outcome obtained from 11 studies was positive observations of changes to the geographical lung area, sound energy or both, while positive observation of lung consolidation has been reported in the remaining 4 studies.

-2 different modalities of lung assessment were used in 8 studies, with 1 study comparing VRI against chest Xray, 1 study comparing VRI with LUS, 2 studies comparing LUS to computed tomography (CT) and 4 studies comparing LUS in contrast to CT

3- I think, the authors should not use abbreviations at the beginning of sentences and lines (e.g.) High-frequency chest wall oscillation………

- HFCWO device has been enhanced over the year such as integrating electronic control for specified pressure oscillating discs to deliver palpitation directly to targeted chest areas [6], [7].

-LUS demonstrated statistical equivalent to chest X-ray in detecting respiratory diseases such as lung consolidation and pleural effusion from 50 patients’ results, in terms of sensitivity [33].

-VRI has been proposed to monitor respiratory distribution within the lungs dynamically and is regarded as an electronic stethoscope alternative which records vibration emitted from the chest using an array of microphones and converts them into grey-scale images [29], [44], [45].

4- I think, the authors should not use abbreviation in title and subtitle (e.g.) Lung ultrasound, Vibration response imaging, computed tomography

3.4.1. LUS

3.4.2. LUS Against Chest X-rays

3.4.3 LUS Against CT

3.4.4. VRI

3.4.5. VRI Against Chest X-rays

3.4.6. VRI Against LUS

4- I think, the authors should not use abbreviation in title and subtitle (e.g.) Lung ultrasound, Vibration response imaging, computed tomography

3.4.1. LUS

3.4.2. LUS Against Chest X-rays

3.4.3 LUS Against CT

3.4.4. VRI

3.4.5. VRI Against Chest X-rays

3.4.6. VRI Against LUS

Reviewer 5 Report

Very interesting topic and highly topical for potential readers of the Journal. Excellent work, congratulations! However, some comments are made in favor of improving the current version of the manuscript.

.- Abstract. OK.

.- Keywords. Maybe delete “biosensing”.

.- Introduction. OK.

.- Methodology. OK.

.- Results. Regarding the delimitation of bibliographic citations. Instead of “[26], [27], [28], [29], [30], [31]”, it could be expressed as “[26-31]”.

.- Discussion. Adequate in form and substance.

.- Conclusions. Perhaps the bibliographic citations could be removed. Not be usual in this section.

.- Figures and tables. Figure 1, okay. Tables 1 and 2, perhaps the "N" of the included studies could be added in the title. Table 3, very accurate.

.- References. Of the 46 citations, only 15 (33%) are recent (that is, five years or less from their publication). Assess including any recent additional citation.
